# EXCESS RISK ANALYSIS FOR EPISTEMIC UNCERTAINTY WITH APPLICATION TO VARIATIONAL INFERENCE

## ABSTRACT

Bayesian deep learning plays an important role especially for its ability evaluating epistemic uncertainty (EU). Due to computational complexity issues, approximation methods such as variational inference (VI) have been used in practice to obtain posterior distributions and their generalization abilities have been analyzed extensively, for example, by PAC-Bayesian theory; however, little analysis exists on EU, although many numerical experiments have been conducted on it. In this study, we analyze the EU of supervised learning in approximate Bayesian inference by focusing on its excess risk. First, we theoretically show the novel relations between generalization error and the widely used EU measurements, such as the variance and mutual information of predictive distribution, and derive their convergence behaviors. Next, we clarify how the objective function of VI regularizes the EU. With this analysis, we propose a new objective function for VI that directly controls the prediction performance and the EU based on the PAC-Bayesian theory. Numerical experiments show that our algorithm significantly improves the EU evaluation over the existing VI methods.

## 1 INTRODUCTION

As machine learning applications spread, understanding the uncertainty of predictions is becoming more important to increase our confidence in machine learning algorithms (Bhatt et al., 2021). Uncertainty refers to the variability of a prediction caused by missing information. For example, in regression problems, it corresponds to the error bars in predictions; and in classification problems, it is often expressed as the class posterior probability, entropy, and mutual information (Hüllermeier & Waegeman, 2021; Gawlikowski et al., 2022). There are two types of uncertainty (Bhatt et al., 2021): 1) Aleatoric uncertainty (AU), which is caused by noise in the data itself, and 2) Epistemic uncertainty (EU), which is caused by a lack of training data. In particular, since EU can tell us where in the input space is yet to be learned, integrated with deep learning methods, it is used in such applications as dataset shift (Ovadia et al., 2019), adversarial data detection (Ye & Zhu, 2018), active learning (Houlsby et al., 2011), Bayesian optimization (Hernández-Lobato et al., 2014), and reinforcement learning (Janz et al., 2019).

Mathematically, AU is defined as Bayes risk, which expresses the fundamental difficulty of learning problems (Depeweg et al., 2018; Jain et al., 2021; Xu, 2020). For EU, Bayesian inference is useful because posterior distribution updated from prior distribution can represent a lack of data (Hüllermeier & Waegeman, 2021). In practice, measurements like the variance of the posterior predictive distribution, and associated conditional mutual information represented EU in practice (Kendall & Gal, 2017; Depeweg et al., 2018).

In Bayesian inference, since posterior distribution is characterized by the training data and the model using Bayes' formula, its prediction performance and EU are determined automatically. However, due to computational issues, such exact Bayesian inference is difficult to implement; we often use approximation methods, such as variational inference (VI) (Bishop, 2006), especially for deep Bayesian models. Since the derived posterior distribution also depends on the properties of approximation methods, the prediction performance and EU of deep Bayesian learning are no longer automatically guaranteed through Bayes' formula. The prediction performance has been analyzed as generalization error, for example, by PAC-Bayesian theory (Alquier, 2021). Since EU is also essential

in practice, we must obtain a theoretical guarantee of the algorithm- and the sample- dependent non-asymptotic theory for EU, similarly to generalization error analysis.

Unfortunately, study has been limited in that direction. Traditional EU analysis has focused on the properties of the exact Bayesian posterior and predictive distributions (Fiedler et al., 2021; Lederer et al., 2019) as well as large sample behaviors (Clarke & Barron, 1990). Since Bayesian deep learning uses approximate posterior distributions, we cannot apply such traditional EU analysis based on Bayes' formula to Bayesian deep learning. The asymptotic theory of a sufficiently large sample may overlook an important property of the EU that is due to the lack of training data.

Recently, analysis of EU focusing on loss functions was proposed for supervised learning (Xu & Raginsky, 2020; Jain et al., 2021). EU was defined as the **excess risk** obtained by subtracting the Bayes risk corresponding to the AU from the total risk. Thus, excess risk implies the loss due to insufficient data when the model is well specified. Although this approach successfully defines EU with loss functions, the following limitation still remains. Xu & Raginsky (2020) assume that the data generating mechanism is already known and that we can precisely evaluate Bayesian posterior and predictive distribution. A correct model is not necessarily a realistic assumption, and the assumption about an exact Bayesian posterior hampers understanding EU in approximation methods.

To address these limitations, it appears reasonable to analyze excess risk under a similar setting as PAC-Bayesian theory and apply it to EU of approximate Bayesian inference. However, as shown in Sec. 2, analyzing excess risk in such a way leads to impractical theoretical results, and the relations between excess risk and the widely used EU measurements remain unclear. This greatly complicates EU analysis. Because of this difficulty, to the best of our knowledge, no research exists on excess risk for EU for approximation methods.

In this paper, we propose a new theoretical analysis for EU that addresses the above limitations of these existing settings. Our contributions are the followings:

- We show non-asymptotic analysis for widely used EU measurements (Theorems 2 and 3). We propose computing the Bayesian excess risk (BER) (Eq. (9)) and show that this excess risk equals to widely used EU measurements. Then we theoretically show the convergence behavior of BER using PAC-Bayesian theory (Eqs. (13)) and (18)).

- Based on theoretical analysis, we give a new interpretation of the existing VI that clarifies how the EU is regularized (Eqs. (19) and (20)). Then we propose a novel algorithm that directly controls the prediction and the EU estimation performance simultaneously based on PAC-Bayesian theory (Eq. (21)). Numerical experiments suggest that our algorithm significantly improves EU evaluation over the existing VI.

## 2 BACKGROUND OF PAC-BAYESIAN THEORY AND EPISTEMIC UNCERTAINTY

Here we introduce preliminaries. Such capital letters as $X$ represent random variables, and such lowercase letters as $x$ represent deterministic values. All the notations are summarized in Appendix A. In Appendix B, we show summary of the settings.

### 2.1 PAC-BAYESIAN THEORY

We consider a supervised setting and denote input-output pairs by $Z = (X, Y) \in \mathcal{Z} := \mathcal{X} \times \mathcal{Y}$. We assume that all the data are i.i.d. from some unknown data-generating distribution $\nu(Z) = \nu(Y|X)\nu(X)$. Learners can access $N$ training data, $\mathbf{Z}^N := (Z_1, \ldots, Z_N)$ with $Z_n := (X_n, Y_n)$, which are generated by $\mathbf{Z}^N \sim \nu(Z)^N$. We express $\nu(Z)^N$ as $\nu(\mathbf{Z}^N)$. We express conditional distribution as $\nu(Y|X = x)$ as $\nu(Y|x)$ for simplicity. We introduce loss function $l : \mathcal{Y} \times \mathcal{A} \to \mathbb{R}$ where $\mathcal{A}$ is an action space. We express loss of action $a \in \mathcal{A}$ and target variable $y$ is written as $l(y, a)$. We introduce a model $f_\theta : \mathcal{X} \to \mathcal{A}$, parameterized by $\theta \in \Theta \subset \mathbb{R}^d$. When we put a prior $p(\theta)$ over $\theta$, the PAC-Bayesian theory (Alquier, 2021; Germain et al., 2016) guarantees the prediction performance by focusing on the average of the loss with respect to posterior distribution $q(\theta|\mathbf{Z}^N) \in \mathcal{Q}$. $\mathcal{Q}$ is a family of distributions and $q(\theta|\mathbf{Z}^N)$ is not restricted to Bayesian posterior distribution. In this work we consider the log loss and the squared loss. For the log loss, we consider model $p(y|x, \theta)$, and the loss is given as $l(y, p(y|x, \theta)) = -\ln p(y|x, \theta)$, where $\mathcal{A}$ is probability distributions. For the squared loss, we use model $f_\theta(x)$ and $l(y, f_\theta(x)) = |y - f_\theta(x)|^2$, where $\mathcal{Y} = \mathcal{A} = \mathbb{R}$.

PAC-Bayesian theory provides a guarantee for the generalization of test error $R^l(Y|X, \mathbf{Z}^N) := \mathbb{E}_{\nu(\mathbf{Z}^N)}\mathbb{E}_{q(\theta|\mathbf{Z}^N)}\mathbb{E}_{\nu(Z)}l(Y, f_\theta(X))$ and training error $\mathbb{E}_{\nu(\mathbf{Z}^N)}r^l(\mathbf{Z}^N) :=$ $\mathbb{E}_{\nu(\mathbf{Z}^N)}\mathbb{E}_{q(\theta|\mathbf{Z}^N)}\frac{1}{N}\sum_{n=1}^{N}l(Y_n, f_\theta(X_n))$. A typical PAC-Bayesian error bound takes form $R^l(Y|X, \mathbf{Z}^N) \leq \mathbb{E}_{\nu(\mathbf{Z}^N)}r^l(\mathbf{Z}^N) + \text{Gen}^l(\mathbf{Z}^N)$. $\text{Gen}^l(\mathbf{Z}^N)$ is called **generalization error**. Under suitable assumptions (Alquier, 2021), $\text{Gen}^l(\mathbf{Z}^N)$ is upper-bounded by $\mathcal{O}(1/N^\alpha)$ for $\alpha \in (1/2, 1]$. In many cases, it depends on the complexity of the posterior distribution, such as Kullback-Leibler (KL) divergence $\text{KL}(q(\theta|\mathbf{Z}^N)|p(\theta))$. When $\text{Gen}^l(\mathbf{Z}^N) = \text{KL}(q(\theta|\mathbf{Z}^N)|p(\theta))/\lambda + \text{c}$, where $\lambda$ and $c$ are positive constants, given training data $\mathbf{Z}^N = \mathbf{z}^N$, we get a posterior distribution for the prediction by

$$\hat{q}(\theta|\mathbf{z}^N) = \underset{q(\theta|\mathbf{z}^N)\in\mathcal{Q}}{\arg\min} \; r(\mathbf{z}^N) + \frac{\text{KL}(q(\theta|\mathbf{z}^N)|p(\theta))}{\lambda}. \tag{1}$$

When the log loss and $\lambda = N$ is used, this minimization is closely related to variational inference (VI) in Bayesian inference. See (Germain et al., 2016) for details.

Under additional moderate assumptions, using $\hat{q}(\theta|\mathbf{z}^N)$ for the test error, we can derive the following **excess risk (ER)** bound from the PAC-Bayesian generalization bound (Alquier, 2021):

$$\text{ER}^l(Y|X, \mathbf{Z}^N, \theta^*) := R^l(Y|X, \mathbf{Z}^N) - R^l(Y|X, \theta^*) \leq C_1\frac{\ln N}{N^\alpha}, \tag{2}$$

where $R^l(Y|X, \theta^*) = \mathbb{E}_{\nu(Z)}l(Y, f_{\theta^*}(X))$ and $\theta^* = \text{argmin}_\theta\mathbb{E}_{\nu(Z)}l(Y, f_\theta(X))$. Constant $C_1$ depends only on the problem. Since we aim to analyze EU, we do not further discuss the details of the PAC-Bayesian bound. See Appendix C.2 for the explicit conditions of this bound.

Although PAC-Bayesian theory focuses on the average test error over posterior distribution, we use predictive distribution for predictions in Bayesian inference. Thus we define **Prediction Risk (PR)**:

$$\text{PR}^l(Y|X, \mathbf{Z}^N) := \mathbb{E}_{\nu(\mathbf{Z}^N)}\mathbb{E}_{\nu(Z)}l(Y, \mathbb{E}_{q(\theta|\mathbf{Z}^N)}f_\theta(X)). \tag{3}$$

When the loss is log loss, $\text{PR}^{\log}(Y|X, \mathbf{Z}^N) = -\mathbb{E}_{\nu(\mathbf{Z}^N)}\mathbb{E}_{\nu(Z)}\log p^q(Y|X, \mathbf{Z}^N)$ where $p^q(y|x, \mathbf{z}^N) := \mathbb{E}_{q(\theta|\mathbf{z}^N)}p(y|x, \theta)$ is the approximate predictive distribution. Thus, $\text{PR}^{\log}(Y|X, \mathbf{Z}^N)$ corresponds to the log loss of the predictive distribution, which is commonly used in the analysis of Bayesian inference (Watanabe, 2009; 2018).

## 2.2 EPISTEMIC UNCERTAINTY MEASUREMENTS

Here, we introduce widely used EU measurements in approximate Bayesian inference. For the log loss, conditioned on $(X, \mathbf{Z}^N) = (x, \mathbf{z}^N)$, the approximate mutual information has been widely used for uncertainty estimation (Depeweg et al., 2018):

$$I_\nu(\theta; Y|x, \mathbf{z}^N) = H[p^q(Y|x, \mathbf{z}^N)] - \mathbb{E}_{q(\theta|\mathbf{z}^N)}H[p(Y|x, \theta)], \tag{4}$$

where $H[p^q(Y|x, \mathbf{z}^N)] := -\mathbb{E}_{p^q(Y|x, \mathbf{z}^N)}\log p^q(Y|x, \mathbf{z}^N)$ is the entropy of the approximate predictive distribution and $\mathbb{E}_{q(\theta|\mathbf{z}^N)}H[p(Y|x, \theta)]$ is the conditional entropy. $I_\nu(\theta; Y|x, \mathbf{z}^N)$ has been used in Bayesian experimental design (Foster et al., 2019) and reinforcement learning (Depeweg et al., 2018). Note that by taking the expectation, we have $I_\nu(\theta; Y|X, \mathbf{Z}^N) = \mathbb{E}_{\nu(X=x)}I_{\nu(\mathbf{Z}^N=\mathbf{z}^N)}(\theta; Y|x, \mathbf{z}^N)$.

In the case of squared loss, the variance of the model is often used for EU. This is a common practice in VI, Monte Carlo (MC) dropout (Kendall & Gal, 2017), and deep ensemble methods (Lakshminarayanan et al., 2017). Conditioned on $(X, \mathbf{Z}^N) = (x, \mathbf{z}^N)$, it is written as

$$\text{Var}_{\theta|\mathbf{z}^N}f_\theta(x) = \mathbb{E}_{q(\theta|\mathbf{z}^N)}(f_\theta(x) - \mathbb{E}_{q(\theta|\mathbf{z}^N)}f_\theta(x))^2. \tag{5}$$

Although Eqs. (4) and (5) are widely used in application, there have been limited theoretical study for them as discussed in Sec. 1.

## 2.3 EXCESS RISK ANALYSIS AND EPISTEMIC UNCERTAINTY

Recently, the analysis of EU based on excess risk was proposed (Xu & Raginsky, 2020). The key idea of this analysis is to assume that our statistical model $p(y|x, \theta)$ is correct and address the average performance of this model by assuming a prior distribution over $\theta$ with distribution $p(\theta)$.

Specifically, the joint distribution of the training data, the test data, and parameter of the model is given as $p_B(\mathbf{Z}^N, Z, \theta) := p(\theta) \prod_{n=1}^N p(Y_n|X_n, \theta)\nu(X_n)p(Y|X, \theta)\nu(X)$. Under this setting, the goal of learning is to infer decision rule $\psi : \mathcal{Z}^N \times \mathcal{X} \to \mathcal{A}$ that minimizes expected loss $\mathbb{E}_{p_B(\mathbf{Z}^N, Z, \theta)}[l(Y, \psi(X, \mathbf{Z}^N))]$. They refer to this setting as Bayesian learning since we marginalize out parameter $\theta$. With this notation, Xu & Raginsky (2020) defined **minimum excess risk** as

$$\mathrm{MER}^l(Y|X, \mathbf{Z}^N) := \inf_{\psi:\mathcal{Z}^N \times \mathcal{X} \to \mathcal{A}} \mathbb{E}_{p_B(\mathbf{Z}^N, Z, \theta)}[l(Y, \psi(X, \mathbf{Z}^N))] - \inf_{\phi:\Theta \times \mathcal{X} \to \mathcal{A}} \mathbb{E}_{p_B(\mathbf{Z}^N, Z, \theta)}[l(Y, \phi(\theta, X))], \quad (6)$$

where the first term is the minimum achievable risk using the training data and the second term is the Bayes risk since it uses learning rule $\phi : \Theta \times \mathcal{X} \to \mathcal{A}$, which takes true parameter $\theta$ instead of the training data. Thus, the second term is the aleatroic uncertainty (AU) since it expresses the task's fundamental difficulty. Then $\mathrm{MER}$ can be regarded as the EU since it is the difference between the total risk and the AU (Xu & Raginsky, 2020; Hafez-Kolahi et al., 2021).

For the log loss, the first term is $H[p(Y|X, \mathbf{Z}^N)]$ and the second term is $\mathbb{E}_{p(\theta)}H[p(Y|X, \theta)]$. Thus, $\mathrm{MER}^{\log}(Y|X, \mathbf{Z}^N) = I(\theta; Y|X, \mathbf{Z}^N)$, which is the conditional mutual information. Other than the log loss, if the loss function satisfies the $\sigma^2$ sub-Gaussian property conditioned on $(X, \mathbf{Z}^N) = (x, \mathbf{z}^N)$, $\mathrm{MER}^l(Y|X, \mathbf{Z}^N) \le \sqrt{2\sigma^2 I(\theta; Y|X, \mathbf{Z}^N)}$ holds (Xu & Raginsky, 2020). In many practical settings $I(\theta; Y|X, \mathbf{Z}^N)$ is upper-bounded by $\mathcal{O}(\ln N/N)$. Thus, EU converges with $\mathcal{O}(\ln N/N)$ under this settings. See Appendix C for more details.

Although this analysis successfully defined EU with rigorous theoretical analysis, the assumptions are clearly impractical since we assume that the correct model, exact Bayesian posterior, and predictive distributions are available. To extend this analysis into approximate Bayesian inference, it is tempting to combine the theory of MER with PAC-Bayesian theory where the data are generated i.i.d from $\nu(Z)$. For that extension, here we introduce the **Prediction Excess Risk (PER)** using Eq. (3):

$$\mathrm{PER}^l(Y|X, \mathbf{Z}^N) := \mathrm{PR}^l(Y|X, \mathbf{Z}^N) - \inf_{\tilde{\phi}:\mathcal{X} \to \mathcal{A}} \mathbb{E}_{\nu(Z)}[l(Y, \tilde{\phi}(X))], \quad (7)$$

where the second term corresponds to the Bayes risk. Although we introduced this definition inspired by Eq. (6), it is impractical for evaluating EU. In practice, we are interested in evaluating EU using only input $x$, as shown in Eqs. (4) and (5). However, we cannot use Eq. (7) for that purpose since we do not know both $\nu(Y|x)$ and the Bayes risk in the second term. Despite less practical definition, as shown in Sec.3, PER plays a fundamental role in understanding the algorithm-dependent behavior of the widely used EU measurements in Eqs. (4) and (5).

## 3    ANALYSIS OF EPISTEMIC UNCERTAINTY BASED ON EXCESS RISK

In this section, we develop theories for analyzing the widely used EU measurements introduced in Sec 2.2. We focus on the following questions. **(Q1)** The convergence behaviors of those measurements are not apparent. As the number of training data points increases, we expect these measurements to converge to zero. **(Q2)** The relationship between these measurements and the generalization error is unclear. Since these measurements depend on the training data and the algorithm, we expect some meaningful relationships must exist. All the proofs in this section are shown in Appendix D.

### 3.1    RELATION BETWEEN EPISTEMIC UNCERTAINTY AND BAYESIAN EXCESS RISK

First, to connect the practical EU evaluation methods in Sec.2.2 with the excess risk analysis in Sec.2.3, we introduce the approximate joint distribution of test data, training data, and parameters:

$$\nu(\mathbf{Z}^N)q(\theta|\mathbf{Z}^N)\nu(Z) \approx p^q(\theta, \mathbf{Z}^N, Z) := \nu(\mathbf{Z}^N)q(\theta|\mathbf{Z}^N)\nu(X)p(Y|X, \theta). \quad (8)$$

When the log loss is used, we employ model for $p(y|x, \theta)$ in Eq. (8). When the squared loss is used, we assume Gaussian distribution $p(y|x, \theta) = N(y|f_\theta(x), v^2)$ for some $v \in \mathbb{R}$.

When a model is well specified, that is, $\nu(y|x) = p(y|x, \theta^*)$ holds for some $\theta^* \in \Theta$, we expect that the predictive distribution converges to $p(y|x, \theta^*)$ and the approximation of Eq. (8) becomes accurate as $N$ increases. We discuss the quality of this approximation in Sec.3.2. Under this setting, we define **Bayesian Excess risk (BER)**:

$$\mathrm{BER}^l(Y|X, \mathbf{Z}^N) := \mathrm{BPR}^l(Y|X, \mathbf{Z}^N) - \inf_{\phi:\Theta \times \mathcal{X} \to \mathcal{A}} \mathbb{E}_{p^q(\theta, \mathbf{Z}^N, Z)}l(Y, \phi(\theta, X)) \quad (9)$$

where BPR is the **Bayesian Prediction Risk** defined as

$$\mathrm{BPR}^l(Y|X, \mathbf{Z}^N) := \mathbb{E}_{p^q(\theta, \mathbf{Z}^N, Z)} l(Y, \mathbb{E}_{q(\theta'|\mathbf{Z}^N)} f_{\theta'}(X)), \tag{10}$$

and the second term is the Bayes risk under the approximate joint distribution of Eq. (8). Note that $\mathrm{BER}^l(Y|X, \mathbf{Z}^N)$ is always larger than 0; see Appendix D.2 for details. We also show the formal definitions of BER and BPR conditioned on $(X, \mathbf{Z}^N) = (x, \mathbf{z}^N)$ in Appendix D.1.

BER and BPR are defined, motivated by PER, PR, and MER. The difference is the mechanism of through which the test data are generated. In BER, we assume that our model $p(y|x, \theta)$ is correct, and the parameters follow the approximate posterior distribution $q(\theta|\mathbf{z}^N)$. Thus, the data-generating mechanism resembles the setting in Sec.2.3. Therefore, similar to MER, BER implies the loss due to insufficient data under the assumption that our current model $q(\theta|\mathbf{z}^N)p(y|x, \theta)$ is correct.

The next theorem elaborate this intuition and connects BER to widely used EU measurements:

**Theorem 1.** *Conditioned on $(x, \mathbf{z}^N)$, we express $\mathrm{BER}^{\log}(Y|x, \mathbf{z}^N)$ for the log loss and $\mathrm{BER}^{(2)}(Y|x, \mathbf{z}^N)$ for the squared loss. Under the definition of Eq. (9), we have*

$$\mathrm{BER}^{\log}(Y|x, \mathbf{z}^N) = I_\nu(\theta; Y|x, \mathbf{z}^N), \qquad \mathrm{BER}^{(2)}(Y|x, \mathbf{z}^N) = \mathrm{Var}_{\theta|\mathbf{z}^N} f_\theta(x). \tag{11}$$

Thus, by studying BER, we can analyze the widely used EU measurements. We point out that $\mathrm{BPR}^{\log}(Y|x, \mathbf{z}^N) = H[p^q(Y|x, \mathbf{z}^N)]$, and the Bayes risk is given as $\mathbb{E}_{q(\theta|\mathbf{z}^N)} H[p(Y|x, \theta)]$.

**Remark 1.** *BER captures EU when our model is well specified. On the other hand, PER (Eq. (7)) represents the prediction performance, which has the relation to the quality of approximation of Eq. (8) under the given loss function. This intuition leads to our new VI in Sec.3.3.*

## 3.2 ANALYSIS OF EXCESS RISK BASED ON PAC-BAYESIAN THEORY

Based on the definitions introduced in Sec. 3.1, here we develop a novel relation between BER and the generalization. First, we show the results of the squared loss. For simplicity, assume $\mathcal{Y} = \mathbb{R}$. See Appendix D.6 for $\mathcal{Y} = \mathbb{R}^d$.

**Theorem 2.** *Conditioned on $(x, \mathbf{z}^N)$, assume that a regression function is well specified, that is, $\mathbb{E}_{\nu(Y|x)}[Y|x] = f_{\theta^*}(x)$ holds. Then we have*

$$\mathrm{PER}^{(2)}(Y|x, \mathbf{z}^N) + \mathrm{BER}^{(2)}(Y|x, \mathbf{z}^N) = \mathrm{ER}^{(2)}(Y|x, \mathbf{z}^N, \theta^*) \leq R^{(2)}(Y|x, \mathbf{z}^N). \tag{12}$$

*Furthermore assume that the PAC-Bayesian bound Eq. (2) holds, and then we have*

$$\mathrm{PER}^{(2)}(Y|X, \mathbf{Z}^N) + \mathrm{BER}^{(2)}(Y|X, \mathbf{Z}^N) = \mathrm{ER}^{(2)}(Y|X, \mathbf{Z}^N, \theta^*) \leq \frac{C_1 \ln N}{N^\alpha}. \tag{13}$$

*Proof.* We use the following relation about the Jensen gap and BER:

**Lemma 1.** *For any $(x, y)$ and any posterior distribution conditioned on $\mathbf{Z}^N = \mathbf{z}^N$, we have*

$$|y - \mathbb{E}_{q(\theta|\mathbf{z}^N)} f_\theta(x)|^2 + \mathrm{Var}_{\theta|\mathbf{z}^N} f_\theta(x) = \mathbb{E}_{q(\theta|\mathbf{z}^N)} |y - f_\theta(x)|^2. \tag{14}$$

From this lemma, the theorem follows directly. □

**Remark 2.** *When we use a flexible model, such as a deep neural network for $f_\theta(x)$, assumption $\mathbb{E}_{\nu(Y|x)}[Y|x] = f_{\theta^*}(x)$ holds even when $\nu(Y|x) \neq p(y|x, \theta^*)$, which means we misspecify the noise function in regression tasks.*

From Eq. (12), $\mathrm{Var}_{\theta|\mathbf{z}^N} f_\theta(x)$ clearly is a lower bound of excess risk and test error, consistent with the well-known result that the variance of the predictor often underestimates EU (Lakshminarayanan et al., 2017). From Eq. (13), $\mathrm{BER}^{(2)}(Y|X, \mathbf{Z}^N)$ converges to 0 with the same order as the PAC-Bayesian bound. Finally, we remark that from Lemma 1, we have

$$R^{(2)}(Y|X, \mathbf{Z}^N) = \mathrm{PR}^{(2)}(Y|X, \mathbf{Z}^N) + \mathrm{BER}^{(2)}(Y|X, \mathbf{Z}^N). \tag{15}$$

This indicates that the test error is decomposed into PR and BER. As pointed out in Remark 1, BER is EU under the approximation of Eq. (8), and PER represents the quality of that approximation, Eq. (15) suggests that the test error simultaneously regularizes those BER and PER.

Next we show the log loss result. Our analysis requires additional assumption about model $p(y|x, \theta)$. We define log density ratio $L(y, x, \theta, \theta^*) := -\ln p(y|x, \theta) + \ln p(y|x, \theta^*)$.

**Assumption 1.** *Conditioned on $(x, \theta, \mathbf{z}^N)$, there exists convex function $h(\rho)$ for $[0, b)$ such that cumulant function $L(y, x, \theta, \theta^*)$ is upper-bounded by $h(\rho)$, i.e., the following inequality holds:*

$$\ln \mathbb{E}_{p(Y|x,\theta)} e^{\rho(L(Y,x,\theta,\theta^*) - \mathbb{E}_{p(Y|x,\theta)} L(Y,x,\theta,\theta^*))} \le h(\rho). \tag{16}$$

For example, if $h(\rho) = \rho^2 \sigma^2(x, \theta)/2$ and $b = \infty$, this assumption resembles the $\sigma^2$ sub-Gaussian property given $(x, \theta, \mathbf{z}^N)$. When considering Gaussian likelihood $p(y|x, \theta) = N(y|f_\theta(x), v^2)$, we have $h(\rho) = \frac{\rho^2}{2v^2}|f_\theta(x) - f_{\theta*}(x)|^2$. Thus, $\sigma^2(x, \theta)$ depends on $x$ and $\theta$, and we refer to this $\sigma^2(x, \theta)$ as a sub-Gaussian property. Other than the Gaussian likelihood, when the log loss is bounded, it satisfies the sub-Gaussian property. In this paper, we focus on this sub-Gaussian setting for Assumption 1 to clarify the presentation. We show an example of the logistic regression in Appendix D.11.

**Theorem 3.** *When the model is well specified, that is, $\nu(y|x) = p(y|x, \theta^*)$ holds and $\sigma^2(x, \theta)$ sub-Gaussian property is satisfied for $L(y, x, \theta, \theta^*)$, as discussed above. Assume that $\mathbb{E}_{q(\theta|\mathbf{z}^N)} \sigma^2(x, \theta) < \sigma_p^2 < \infty$. Conditioned on $(x, \mathbf{z}^N)$, we have*

$$\text{PER}^{\log}(Y|x, \mathbf{z}^N) + \text{BER}^{\log}(Y|x, \mathbf{z}^N) \le \sqrt{2\sigma_p^2 \text{ER}^{\log}(Y|x, \mathbf{z}^N, \theta^*)}. \tag{17}$$

*Moreover, assume PAC-Bayesian bound Eq. (2) and $\mathbb{E}_{\nu(\mathbf{z}^N)q(\theta|\mathbf{Z}^N)\nu(X)} \sigma^2(X, \theta) < \sigma_q^2 < \infty$ hold, and then we have*

$$\text{PER}^{\log}(Y|X, \mathbf{Z}^N) + \text{BER}^{\log}(Y|X, \mathbf{Z}^N) \le \sqrt{2\sigma_q^2 \text{ER}^{\log}(Y|X, \mathbf{Z}^N, \theta^*)} \le \sqrt{\frac{2\sigma_q^2 C_1 \ln N}{N^\alpha}}. \tag{18}$$

*Proof.* Using Assumption 1, we apply change-of-measure inequalities (Ohnishi & Honorio, 2021) to control the approximation error of Eq. (8). □

**Remark 3.** *When we use a generalized linear model, the assumption of a well-specified model is relaxed so that $\mathbb{E}_{\nu(Y|x)}[Y|x]$ is well specified, similar to Remark 2. See Appendix D.9 for details.*

From Eq. (17), $I_\nu(\theta; Y|x, \mathbf{z}^N)$ is a lower bound of the excess risk and test error. From Eq. (18), $I_\nu(\theta; Y|X, \mathbf{Z}^N)$ converges in the order of $\mathcal{O}(\sqrt{\ln N/N^\alpha})$ if we can upper-bound $\mathbb{E}_{\nu(\mathbf{Z}^N)q(\theta|\mathbf{Z}^N)\nu(X)} \sigma^2(X, \theta) < \sigma_q^2 < \infty$. For the Gaussian likelihood, we have $\mathbb{E}_{\nu(\mathbf{Z}^N)q(\theta|\mathbf{Z}^N)\nu(X)} \sigma^2(X, \theta) \le 2\text{ER}^{\log}(Y|X, \mathbf{Z}^N, \theta^*) \le \frac{2C_1 \ln N}{N^\alpha} := \sigma_q^2$. See Appendix D.8 for a detail.

In a similar way, we can derive the convergence rate of the entropy of the predictive distribution, which shows $H[p^q(Y|X, \mathbf{Z}^N)] = H[p(Y|X, \theta^*)] + \mathcal{O}(\sqrt{\ln N/N^\alpha})$. See Appendix D.10 for a formal statement. We show similar results for Theorem 3 under sub-exponential property in Appendix E.

In summary, we obtained the convergence of widely used EU measurements and the entropy in the approximate Bayesian inference for the first time. They converges faster than excess risks. Moreover we obtained two messages from Theorems 2 and 3. First, the widely used EU measurements are the lower bounds of the test error and excess risks. This is consistent with the experimental fact that these EU measurements often underestimate EU. Second, the sum of PER and BER is upper-bounded by excess risk (test error). Thus, when minimizing the test error, we also simultaneously minimize PER and BER. This interpretation extends the intuition of Remark 1 and leads to a new VI in Sec. 3.3.

## 3.3 NOVEL EU REGULARIZATION METHOD FOR VARIATIONAL INFERENCE

As seen in Sec. 3.2, minimizing the test error leads to minimizing BER and PER. In this section, we discuss this relation using the objective function of VI. As an explicit example, consider a regression problem using $N(y|f_\theta(x), v^2)$. Then from Lemma 1, the loss function of the standard VI (eliminating $\text{KL}(q(\theta|\mathbf{z}^N)|p(\theta))$) can be written:

$$-\mathbb{E}_{\nu(Z)}\mathbb{E}_{q(\theta|\mathbf{z}^N)} \ln N(y|f_\theta(x), v^2) = \mathbb{E}_{\nu(Z)} \frac{|Y - \mathbb{E}_{q(\theta|\mathbf{z}^N)}f_\theta(X)|^2 + \text{Var}_{\theta|\mathbf{z}^N} f_\theta(X)}{2v^2} + \frac{\ln 2\pi v^2}{2}$$

$$= \frac{\text{PR}^{(2)}(Y|X, \mathbf{z}^N) + \text{BER}^{(2)}(Y|X, \mathbf{z}^N)}{2v^2} + \frac{\ln 2\pi v^2}{2}. \tag{19}$$

Eq. (19) implies that the standard VI tries to fit the mean of the predictive distribution to target variable $y$ with the regularization term about the variance of the predictor. These terms corresponds

to $\mathrm{PR}^{(2)}(Y|X,\mathbf{z}^N)$ and $\mathrm{BER}^{(2)}(Y|X,\mathbf{z}^N)$. Note that there is a relation $\mathrm{BER}^{\log}(Y|x,\mathbf{z}^N) \leq \mathrm{Var}_{\theta|\mathbf{z}^N} f_\theta(x)/v^2$ for the Gaussian likelihood. This interpretation is consistent with Remark 1.

It has been numerically reported that the standard VI often underestimates EU. Alternative objective functions have been proposed to address this issue. For example, the entropic loss defined as $\mathrm{Ent}_\alpha^l(y,x) := -\frac{1}{\alpha} \ln \mathbb{E}_{q(\theta|\mathbf{z}^N)} e^{-\alpha l(y,f_\theta(x))}$ for $\alpha > 0$, which is used in the $\alpha$-divergence dropout ($\alpha$-DO) (Li & Gal, 2017) and the second order PAC-Bayesian methods ($2^{\mathrm{nd}}$-PAC) (Masegosa, 2020; Futami et al., 2021), can capture EU better than the standard VI. Note that when $\alpha = 1$ and the log loss is used, the entropic risk corresponds to the log loss using the predictive distribution. For the Gaussian likelihood, we can upper-bound the entropic risk:

$$\mathbb{E}_{\nu(Z)}\mathrm{Ent}_{\alpha=1}^{\log}(Y,X) \leq \mathbb{E}_{\nu(Z)}\frac{|Y-\mathbb{E}_{q(\theta|\mathbf{z}^N)}f_\theta(X)|^2}{v^2}+\frac{\mathrm{Var}_{\theta|\mathbf{z}^N}f_\theta(X)}{v^2}-\mathrm{BER}^{\log}(Y|X,\mathbf{z}^N)+\ln 2\pi v^2, \quad (20)$$

where we used Eq. (17). See Appendix D.12 for the derivation. Compared to Eq. (19), the entropic risk implicitly introduces a smaller regularization term for BER. This explains why $\alpha$-DO and $2^{\mathrm{nd}}$-PAC showed larger EU than the standard VI. We show a similar result for the entropic risk of the general log loss other than the Gaussian likelihood in Appendix D.12.

From these relations, balancing BER and PR appropriately leads to a solution that better evaluates the EU. Motivated by the decomposition in Eqs. (19) and (20), we directly control the prediction performance and the Bayesian excess risk for the Gaussian likelihood:

$$\mathrm{rBER}(\lambda) = \frac{1}{N}\sum_{i=1}^N \frac{|y_i-\mathbb{E}_{q(\theta|\mathbf{z}^N)}f_\theta(x_i)|^2}{2v^2}+\lambda\frac{\mathrm{Var}_{\theta|\mathbf{z}^N}f_\theta(x_i)}{2v^2}+\frac{\ln 2\pi v^2}{2}+\frac{1}{N}\mathrm{KL}(q(\theta|\mathbf{z}^N)|p(\theta)), \quad (21)$$

where $0 < \lambda \leq 1$ is the coefficient of the BER regularizer. $\mathrm{KL}(q(\theta|\mathbf{z}^N)|p(\theta))$ is a regularization term motivated by PAC-Bayesian theory. We select $\lambda$ by cross-validation and it should be smaller than 1 since $\lambda = 1$ corresponds to the standard VI from Eq. (19) and the standard VI often underestimates the EU. We call Eq. (21) the regularized Bayesian Excess Risk VI (rBER) and show the PAC-Bayesian generalization guarantee for our rBER in Appendix F. In Sec. 5.2, we numerically evaluated this objective function.

rBER can also be seen as an extension of the standard VI. In the standard VI, the test loss is lower-bounded by the sum of the PR and BER with equal weights (Eq. (15)). rBER has the flexible weights between PR and BER. See Appendix F for a detailed comparison.

## 4 RELATION TO EXISTING WORK

The existing theoretical analysis of uncertainty focused on the calibration performance and clarified when a model over- and underestimates uncertainty (Nixon et al.; Bai et al., 2021; Naeini et al., 2015; Guo et al., 2017). We did not focus on analyzing the calibration property in this work because considering only the EU is not sufficient to deal with the calibration. We need to study both the AU and EU simultaneously. This is the limitation of our work. Other than calibration, the analysis of Gaussian processes (GP) has been gaining attention since GP's posterior predictive distribution can be expressed analytically (Fiedler et al., 2021; Lederer et al., 2019). Some research focused on the distance or geometry between the test and training data points to derive EU (Liu et al., 2020; Tian et al., 2021). Other approaches connect the randomness of the posterior distribution to predictions by the delta method (Nilsen et al., 2022). Differently, the information-theoretic approach (Xu & Raginsky, 2020) focused on the loss function of the problem and defined the excess risk as the EU. Loss function-based analysis was proposed in the deterministic learning algorithm (Jain et al., 2021). Our theory, which can be regarded as an extension of the information-theoretic approach (Xu & Raginsky, 2020) to approximate Bayesian inference, derived the convergence properties of the variance and the entropy of the posterior predictive distributions.

Although the excess risk bound in Eq. (2) has been discussed by PAC-Bayesian theory (Alquier, 2021), its relation to the EU has not been investigated. The relationship between PAC-Bayesian theory and Bayesian inference has been investigated in terms of marginal likelihood (Germain et al., 2016; Rothfuss et al., 2021). Our work established new relationships that connect the uncertainty of the Bayesian predictive distribution and the PAC-Bayesian generalization bound. The information-theoretic approach (Xu & Raginsky, 2020) clarified that EU can be expressed by conditional mutual information. This relation was extended to meta-learning (Jose et al., 2021). However, the researchers

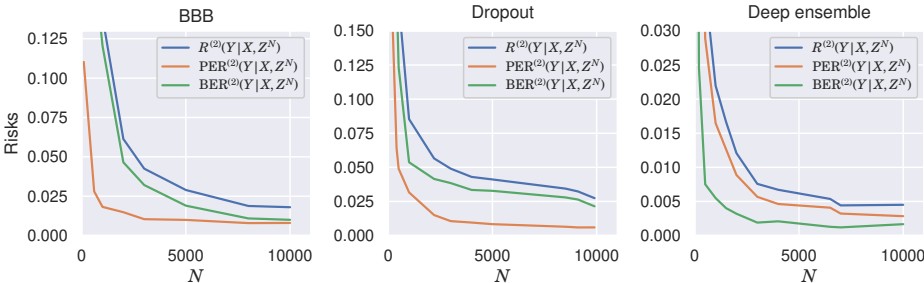

Figure 1: Rresult of toy data experiments: $N$ represents number of training data points, and vertical line is value of each excess risk.

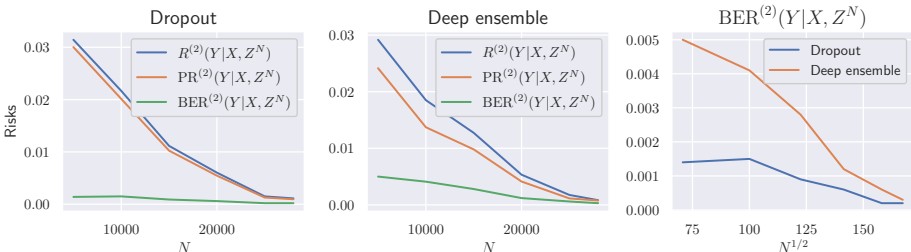

Figure 2: Real data experiments of depth estimation: Vertical line is the value of each risk.

assumed that correct models and exact posterior distributions are available. Our proposed analysis relaxes these assumptions.

## 5 NUMERICAL EXPERIMENTS

In this section, we numerically confirm the theoretical findings in Sec. 3 and our proposed rBER in Eq. (21). We show the detailed experimental settings and additional results in Appendix G.

### 5.1 NUMERICAL EVALUATION OF THEOREM 2

We numerically confirm the statement of Theorem 2. First, we consider toy data experiments where the true model is $y = 0.5x^3 + \epsilon$, $\epsilon \sim N(0,1)$, $x \sim N(0,1)$. We consider a Bayesian neural network (BNN) for $f_\theta(x)$ as a 4 layer neural network model with ReLU activation. We approximate the posterior distribution of the parameters of the neural network by Bayes by back-propagation (BBB)(Hernández-Lobato & Adams, 2015), dropout (Kendall & Gal, 2017), and deep ensemble (Lakshminarayanan et al., 2017). We evaluate $\mathrm{PER}^{(2)}(Y|X,\mathbf{Z})$, $\mathrm{BER}^{(2)}(Y|X,\mathbf{Z})$ $(:= \mathbb{E}_{\nu(X)}\mathrm{Var}_{\theta|\mathbf{Z}^N} f_\theta(X))$, and $R^{(2)}(Y|X,\mathbf{Z})$ (test error). The results are shown in Fig. 1. Our numerical results satisfy Eq. (13) in Theorem 2, that is, $\mathrm{PER}^{(2)}(Y|X,\mathbf{Z})$ and $\mathrm{BER}^{(2)}(Y|X,\mathbf{Z})$ are upper-bounded by $R^{(2)}(Y|X,\mathbf{Z})$ and converge to zero as the number of samples increases. We calculated the Spearman Rank Correlation (SRC) among $\mathrm{PER}^{(2)}(Y|X,\mathbf{Z})$, $\mathrm{BER}^{(2)}(Y|X,\mathbf{Z})$, and $R^{(2)}(Y|X,\mathbf{Z})$ and showed at least 0.97 suggesting high correlation relation between them.

Next, we confirm Theorem 2 using a real-world dataset. Following the setting of existing work (Amini et al., 2020), we trained a U-Net style network (Ronneberger et al., 2015) with the data of the NYU Depth v2 dataset (Silberman et al., 2012), which consists of RGB-to-depth. We applied dropout and deep ensemble methods. Since we cannot evaluate $\mathrm{PER}^{(2)}(Y|X,\mathbf{Z})$, we instead evaluated Eq. (15), which only requires $\mathrm{PR}^{(2)}(Y|X,\mathbf{Z})$, $\mathrm{BER}^{(2)}(Y|X,\mathbf{Z})$, and $R^{(2)}(Y|X,\mathbf{Z})$. The result is shown in Fig. 2. We found that $\mathrm{PR}^{(2)}(Y|X,\mathbf{Z})$, $\mathrm{BER}^{(2)}(Y|X,\mathbf{Z})$ are upper-bounded by $R^{(2)}(Y|X,\mathbf{Z})$ for real dataset experiments. We calculated the SRC among $\mathrm{PR}^{(2)}(Y|X,\mathbf{Z})$, $\mathrm{BER}^{(2)}(Y|X,\mathbf{Z})$, and

Table 1: Benchmark results on test RMSE, PICP, and MPIW.

| Dataset | Avg. Test RMSE | | | | Avg. Test PICP and MPIW in parenthesis | | | |
|---|---|---|---|---|---|---|---|---|
| | f-SVGD | VAR | rBER(0) | rBER(0.05) | f-SVGD | VAR | rBER(0) | rBER(0.05) |
| Concrete | 4.33±0.8 | 4.30±0.7 | 4.47±0.6 | 4.48±0.7 | 0.82±0.03 (0.13±0.00) | 0.87±0.04 (0.16±0.01) | 0.99±0.02 (0.50±0.04) | **0.95±0.02** (0.25±0.02) |
| Boston | 2.54±0.50 | 2.53±0.50 | 2.53±0.50 | 2.53±0.51 | 0.63±0.07 (0.10±0.02) | 0.76±0.05 (0.14±0.01) | 0.97±0.01 (0.33±0.04) | **0.92±0.04**(0.22±0.02) |
| Wine | 0.61±0.04 | 0.61±0.04 | 0.64±0.04 | 0.63±0.02 | 0.79±0.03 (0.32±0.05) | 0.85±0.02 (0.39±0.06) | 0.99±0.00 (1.61±0.00) | **0.95±0.03** (0.32±0.15) |
| Power | 3.78±0.14 | 3.75±0.13 | 3.66±0.15 | 3.69±0.12 | 0.43±0.01 (0.07±0.00) | 0.82±0.01 (0.15±0.00) | 0.99±0.01 (0.81±0.01) | 0.96±0.01 (0.37±0.01) |
| Yacht | 0.64±0.28 | 0.60±0.28 | 0.75±0.41 | 0.78±0.48 | 0.92±0.04 (0.02±0.01) | 0.93±0.04 (0.04±0.01) | **0.96±0.03** (0.10±0.01) | **0.94±0.04** (0.08±0.01) |
| Protein | 3.98±0.54 | 3.92±0.05 | 3.83±0.10 | 3.85±0.05 | 0.53±0.01 (0.24±0.01) | 0.83±0.00 (0.58±0.01) | 1.0 ±0.00 (5.04±0.01) | **0.96±0.01** (0.86±0.00) |

Table 2: Cumulative regret relative to that of the uniform sampling.

| Dataset | MAP | $\text{PAC}_\text{E}^2$ | f-SVGD | VAR | rBER(0) | rBER(0.01) | rBER(0.05) |
|---|---|---|---|---|---|---|---|
| Mushroom | 0.129±0.098 | 0.037±0.012 | 0.043±0.009 | 0.029±0.010 | 0.075±0.005 | **0.024±0.009** | **0.021±0.004** |
| Financial | 0.791±0.219 | 0.189±0.025 | 0.154±0.017 | 0.155±0.024 | 0.351±0.030 | **0.075±0.024** | **0.075±0.031** |
| Statlog | 0.675 ±0.287 | 0.032±0.003 | 0.010±0.000 | 0.006±0.000 | 0.145±0.223 | 0.005±0.001 | **0.005±0.000** |
| CoverType | 0.610±0.051 | 0.396±0.006 | 0.372±0.007 | **0.291±0.004** | 0.610±0.051 | 0.351±0.003 | **0.290±0.002** |

$R^{(2)}(Y|X, \mathbf{Z})$ and showed at least 0.98, suggesting a high correlation relation between them. We also evaluated the convergence behaviors of BER and show the result on the right in Fig. 2. BER converges with $\mathcal{O}(1/N^{1/2})$, which is consistent with Eq. (13).

## 5.2 REAL DATA EXPERIMENTS OF REGULARIZED BAYESIAN EXCESS RISK VI

We numerically compared the prediction and EU evaluation performances of our proposed method shown in Eq. (21) in regression and contextual bandit tasks. Motivated by the success of the entropic risk in particle VI (PVI) (Masegosa, 2020; Futami et al., 2021), which approximates the posterior distribution by the ensemble of models, we also applied our rBER to the PVI setting. Thus, the posterior distribution is expressed as $q(\theta) := \frac{1}{N} \sum_{i=1}^{M} \delta_{\theta_i}(\theta)$, where $\delta_{\theta_i}(\theta)$ is the Dirac distribution that has a mass at $\theta_i$. See Appendix G for details about PVI. We refer to rBER(0) when $\lambda = 0$ in Eq. (21). We compared our method with the existing PVI methods, f-SVGD (Wang et al., 2019), $\text{PAC}_\text{E}^2$ (Masegosa, 2020), and VAR (Futami et al., 2021).

We used the UCI dataset (Dheeru & Karra Taniskidou, 2017) for regression tasks. The model is a single-layer network with ReLU activation, and we used 20 ensembles. The results of 20 repetitions are shown in Table 1. We evaluated the fitting performance by RMSE and the uncertainty estimation performance by the prediction interval coverage probability (PICP), which shows the number of test observations inside the estimated prediction interval where the interval was set to 0.95. PICP is best when it is close to 0.95. We evaluated the mean prediction interval width (MPIW), which shows an average width of a prediction interval. A smaller MPIW is a better uncertainty estimate when PICP is near the best. Due to space limitations, the results of $\text{PAC}_\text{E}^2$ and the other $\lambda$s and the negative log-likelihood are shown in Appendix G. We found the existing PVIs show small PICP and MPIW, indicating that the existing methods underestimate the uncertainty. rBER(0) shows a large PICP and MPIW since the Bayesian excess risk is not regularized. rBER(0.05) shows a moderate MPIW with a better PICP and almost identical prediction performance in RMSE. Thus, rBER successfully controlled the prediction and the uncertainty evaluation performances.

Next we evaluated the rBER using contextual bandit problems (Riquelme et al., 2018). We need to balance the trade-off between exploitation and exploration to achieve small cumulative regret. For that purpose, our algorithms must appropriately control the prediction and uncertainty evaluation performance. We used the Thompson sampling algorithm with BNN and two hidden layers. We used 20 ensembles for approximating the posterior distribution. The results of 10 repetitions are shown in Table 2. Our approach outperformed other methods, which means our proposed method showed better prediction and uncertainty control than the existing methods.

## 6 CONCLUSION

We theoretically and numerically analyzed the epistemic uncertainty of approximate inference. We clarified the novel relations among excess risk, epistemic uncertainty, and generalization error. We then showed the convergence rate of the widely used uncertainty measures for the first time. Motivated by theoretical analysis, we proposed a novel variational inference (VI) and applied it to the particle VI. In future work, it would be interesting to explore the relation between BER and evidential learning.

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
