# OpenReview forum: "Excess risk analysis for epistemic uncertainty with application to variational inference"
_ICLR.cc/2023/Conference — Submitted to ICLR 2023_

### Official Review · Reviewer_6WHt · 2022-10-23

**Confidence:** 5
**Correctness:** 1
**Technical Novelty And Significance:** 2
**Empirical Novelty And Significance:** 3
**Recommendation:** 3

**Clarity, Quality, Novelty And Reproducibility:**

The writing is reasonably clear, although the notations might be optimized.  E.g., it might be better to highlight the dependency of various quantities such as $ER^l$ on $q$.

**Strength And Weaknesses:**

This work studies a very relevant problem: whether (approximate) Bayesian epistemic uncertainty estimates are valid upper bounds for the excess risk, in which case they can be viewed as valid in a frequentist sense.  The authors showed that posterior variance-like quantities can be *lower bounds* for the excess risk, which would be a serious issue and worth addressing.

However, it appears to me that the authors studied the wrong excess risk: they considered the excess risk of the stochastic predictor derived from the Gibbs posterior,
$$
R^l(Y | X, \mathbf{Z}^N) := E_{Z^n} E_{\color{blue}q(\theta\mid Z^n)} E_{Z=(X,Y)} \ell(Y, f_\theta(X)),
$$
as opposed to the deterministic predictor which approximates the true Bayes predictor.  The former predictor only achieves order-optimal excess risk, which is reasonable for studying contraction properties such as predictive error, but usually unreasonable as a target for uncertainty quantification.

As a concrete example, for homoscedastic regression problems with a square loss, our point estimate is typically based on the posterior mean estimate $\hat f_n(x) := \mathbb{E}_{q(\theta\mid Z^n)} f_\theta(x)$, and we are most interested in whether posterior variance quantities such as $\mathbb{E}\_{q(\theta\mid Z^n)}\big(f\_\theta(x) - \hat f_n(x)\big)^2$ upper bounds the excess risk-type quantities for the posterior mean estimator, $\mathbb{E}\_{\nu(Y|X=x)} (\hat f\_n(x) - Y)^2 - \text{(Bayes Risk at $x$)} = (\hat f_n(x) - f_0(x))^2$.  This corresponds to the scenarios where the credible intervals may have correct coverage.  However, the authors in their Theorem 2 compared this posterior variance with the quantity
$$
R^l(Y | x, \mathbf{Z}^N) - \text{(Bayes Risk)} = \underbrace{\mathbb{E}\_{q(\theta\mid Z^n)}\big(f\_\theta(x) - \hat f_n(x)\big)^2}\_{\mathrm{BER}(Y|x,\mathbf{z}^N)} +  \underbrace{((\hat f_n(x) - f_0(x))^2}\_{\mathrm{PER}(Y | x, \mathbf{z}^N)}.
$$
Clearly, the fact that the posterior variance fails to upper bound the above says nothing about its frequentist validity.

As another sanity check, it is also unconvincing that theoretical explanations for the under-coverage issue can be provided in such general settings, as for the results in Section 3: the theorems apply to the true Bayesian posterior and correctly specified (low-dimensional) parametric models.  Clearly, the right conclusion shouldn't be that under-coverage happens across all such settings.


**Summary Of The Paper:**

This works studies the frequentist validity of popular epistemic uncertainty estimates, showing that they equal certain "Bayes excess risk" measures (Thm. 1) and are *lower bound* for the unobservable excess risk of a stochastic predictor derived from the (approximate) posterior.  The authors argue that such behaviors are undesirable and consistent with the fact that epistemic uncertainty are often underestimated in practice.  Based on these observations, the authors proposed a modified variational inference objective for regression problems, where a conditional variance term is down-weighted (Eq. 21).  Empirically the new objective demonstrates improved performance on regression and contextual bandit tasks.

**Summary Of The Review:**

The authors studied a relevant problem, but appeared to have misinterpreted their results.

**Post-rebuttal update**.

I have read the authors' response but did not find them addressing my concerns.  My concerns were twofold:

1. **The messages in Sec 3.2 were misleading.**

For the square loss case, the manuscript appears to draw connections between their Theorem 2, which shows that the posterior variance ($Var_{\theta|z^N} f_\theta(x)$) *lower bounds* their version of excess risk ($ER^{(2)}$), and the experimental fact that the posterior variance underestimates the epistemic uncertainty (EU); as shown in the following quotes:

> From Eq. (12), Varθ|zN fθ(x) clearly is a lower bound of excess risk and test error, consistent with the well-known result that the variance of the predictor often underestimates EU (Lakshminarayanan et al., 2017).

> First, the widely used EU measurements are the lower bounds of the test error and excess risks. This is consistent with the experimental fact that these EU measurements often underestimate EU.

As noted in my review and response, this is only because the authors looked at the wrong definition of ER (for this purpose): for squared loss, it equals
$$
ER^{(2)} = E_{Z^n} E_{q(\theta|Z^n)} E_{Z=(X,Y)} (f_0(X)-f_\theta(X))^2,
$$
as opposed to
$$
E_{Z^n} E_{Z=(X,Y)} (f_0(X) - E_{q(\theta|Z^n)} f_\theta(X))^2.
$$
(The last tuple $(X,Y)$ denote the test input.) The latter is what we want to compare the posterior variance with.

The authors misunderstood my definition of "coverage problem" in the first response, which I have clarified; in their second response, they noted that my concerned relation about $E_{(X,Y)}(Y-E_{q(\theta)} f_\theta(X))^2$ and $E_X E_{q(\theta)} (f_\theta(X) - E_{q(\theta)} f_\theta(X))^2$ (posterior variance) was not analyzed in this work.  But this is precisely my point: what Sec 3.2 analysed was not useful (and misleading), and to explain the experimental fact ofunder-coverage, one should analyze this relation as opposed to the relation studied in Theorem 2.

2. **The motivation of the algorithm was unsatisfactory.**

As the authors noted in their second response, the algorithm is motivated by the fact that "minimizing the standard VI leads to minimizing PR and BER simultaneously, as shown in Eq.~19". (I think they were referring to the standard ELBO.)  Without further explanation, or restriction of the scope of discussion, it is very unclear why this is a problem that needs to be addressed: if we do not restrict the variational family of choice, the true posterior is the minimizer for the ELBO, and clearly it doesn't always suffer from coverage problems.  The remaining of that subsection (3.3) did not offer any convincing explanation.

In summary, while the manuscript contains interesting materials (the algorithm and experiments, for example), it appears that the theoretical discussions need to be thoroughly revised.  Therefore, my rating remains unchanged.

---

> ### Author Response · Authors · 2022-11-16
> **Definition of the excess risk and coverage problems**
>
> Thank you for the comment.
>
> ## Definition of the excess risk
> First, Our excess risk $E^l(Y|x,Z^N):=E_{\nu(Z^n)}E_{q(\theta|Z^N)}E_{Z=X,Y}l(Y,f_\theta(X))$ is not the wrong definition. This excess risk is required to derive the convergence behaviors of the widely used epistemic uncertainty (EU) measurements and their relations to generalization.
>
> The purpose of our analysis is not studying this excess risk. As we introduced in Sec 2, this excess risk has been studied in PAC-Bayesian analysis. The goal of this work is to explore some theoretical properties of the widely used EU measurements in practice, such as the variance of the posterior predictive distribution and associated conditional mutual information, and show how these measurements are related to the objective functions of variational inference.
>
> As for the theoretical properties, as we explaned at the beginning of Sec 3, we are especially interested in
> - 1) the convergence property of the EU and
> - 2) the relation to generalization.
>
> About 1) since the EU represents the loss due to insufficient data, we expect the EU to converge to 0 as the data size increases. About 2), when using an approximate predictive distribution, we want the model to have both a high generalization ability and an adequate EU evaluation performance, and they must have some meaningful relations. In Theorems 1 and 2, we only showed that “the test (excess) risk” always upper bounds the EU, which clarifies the connection between the EU and the generalization error but does not say anything about the coverage problem.
>
> ## Coverage problems
> We did not focus on analyzing the coverage property in this work because considering only the EU is not sufficient to deal with the coverage problem. To study the coverage problem, we need to study both the AU and EU. For example, let us consider constructing the confidence interval of the Bayesian linear regression using the variance of the predictive distribution, which is given as $\sigma^2+\mathrm{Var}f_\theta(x)$ under  model $N(f_\theta(x),\sigma^2)$. As you can see, the second term is the EU, which we analyzed in this work, and the first term corresponds to the noise in the data, so it can be regarded as the AU.
>
> Thus, both the evaluation of AU and EU is required. In this work, since we only discussed the EU,  the coverage problem is the out of scope from this work, and it is the limitation of this work.
> As you pointed out, the coverage problem of the error is an important topic to understand the uncertainty, so currently, we only cite existing work about it in Sec.4. We updated the paper and added the explanation in Sec.4 that the coverage problem is the out of scope and it is the limitation of our analysis.

---

> > ### Comment · Reviewer_6WHt · 2022-11-16
> > **Quick response**
> >
> > Thank you for your response, and I will write a proper update for my review later.  But I want to make a quick clarification about my main concern, around which there seems to be some misunderstanding.
> >
> > My concern regarding coverage refers to the coverage of "epistemic uncertainty estimates", i.e., credible sets for *the regression function $f$*, which can and should be studied separately from "aleatoric uncertainty quantities".  In the least square regression setting, we will construct credible intervals (CIs) for $f(x_0)$, solely based on $Var_{\theta|\mathbf{z}^N} f(x_0;\theta)$ (and the posterior mean estimate $\hat f_n(x_0) := E_{\theta|\mathbf{z}^N}f(x_0;\theta)$).  For such a CI with nominal level $(1-\gamma)$ to have correct coverage in the frequentist sense, we want it to contain $f_0(x_0) = E(\mathbf{y}|\mathbf{x}=x_0)$ w.p. $1-\gamma$, as opposed to containing draws from  $p(\mathbf{y} | \mathbf{x}=x_0)$ w.p. $1-\gamma$.  Only in constructing the predictive intervals of the latter should we involve aleatoric uncertainty-related quantities, such as $\sigma_0^2 = \mathrm{Var}(\mathbf{y}|\mathbf{x}=x_0)$.
> >
> > It is clear from the above that $Var_{\theta|z^N} f(x_0;\theta)$ should only be compared with $(\hat f_n(x_0) - f_0(x_0))^2$. Yet this submission compared similar variance quantities to excess risk quantities which include both $(\hat f_n(x_0) - f_0(x_0))^2$ and $Var(\mathbf{y}|\mathbf{x}=x_0)$ (which equals the Bayes risk), and use the comparison to deduce that such variance quantities (i.e., "epistemic uncertainty estimates") underestimate the epistemic uncertainty.  E.g., on pp.5-6, the submission states:
> >
> > > From Eq. (12), Varθ|zN fθ(x) clearly is a lower bound of excess risk and test error, consistent with the
> > well-known result that the variance of the predictor often underestimates EU (Lakshminarayanan et al.,
> > 2017).
> >
> > > First, the widely used EU measurements are the lower bounds of the test error and excess risks. This is consistent with the experimental fact that these EU measurements often underestimate EU.
> >
> > The claimed consistency (with empirical observations) is clearly incorrect, yet they appear to have motivated the proposed algorithm.  I hope this makes my concern clear.

---

> > > ### Author Response · Authors · 2022-11-17
> > > **The motivation of the proposed algorithm**
> > >
> > > Thank you for the detailed explanation.
> > >
> > > Let us explain the motivation of the proposed algorithm in Sec 3.3. Our algorithm's motivation is not from the coverage problem but rather from how the existing variational inference (VI) algorithms differ in BER. We studied how the existing variational inference methods minimize PR(prediction risk) and BER differently.
> > >
> > > First, we found that minimizing the standard VI leads to minimizing PR and BER simultaneously, as shown in Eq 19. And this relation corresponds to Eq. 12 in Theorem 2.
> > >
> > > On the other hand, as written in the third line on page 7, "It has been numerically reported that the standard VI often underestimates EU. Alternative objective functions have been proposed to address this issue." The alternative objective is the entropic risk used in alpha divergence minimization in VI. And we found that the upper bound of the entropic risk has the negative regularization term about BER, as shown in Eq.20.
> > >
> > > From these results, we considered that balancing BER and PR appropriately leads to a solution that better evaluates the EU, and this leads to the objective function shown in Eq.21. This is an extension of the standard VI and alpha divergence minimization. Thus, in numerical experiments, we compared our algorithm with the standard VI and alpha divergence minimization. Thus, our algorithm is based on the new interpretation of existing VI algorithms using BER.
> > >
> > > The numerical experiments in Fig.1 studied the relation of Theorem2. Fig.1 showed that the test error $R^l(Y|X,ZN):=E_{\nu(Z^N)}E_{q(\theta|Z^N)}E_{\nu(Z)}l(Y,f_\theta(X))$) upper bounds both BER and PER(prediction excess risk). This is the relation presented in Eq 12 in Theorem 2.
> > >
> > > On the other hand,  your concern about the relation: $E_{Y,X}(Y-E_{q(\theta)}f_\theta(X))^2$ and $E_XE_{q(\theta)}(f_\theta(X)-E_{q(\theta)}f_\theta(X))^2)$ is not theoretically analyzed in this work. As shown in Fig.1, for drop out and BBB, $E_{Y,X}(Y-E_{q(\theta)}f_\theta(X))^2 \geq E_XE_{q(\theta)}(f_\theta(X)-E_{q(\theta)}f_\theta(X))^2)$ holds and for deep ensemble, $E_{Y,X}(Y-E_{q(\theta)}f_\theta(X))^2 \leq E_XE_{q(\theta)}(f_\theta(X)-E_{q(\theta)}f_\theta(X))^2)$ holds. So we think it is difficult to derive which is larger in general settings.

---

> ### Author Response · Authors · 2022-12-07
> **Reply to the comment of Post-rebuttal update**
>
> Dear Reviewer
>
> We appreciate your detailed comments. We answer your concerns below:
>
> #### **1 )  The messages in Sec 3.2 were misleading**
> First, we will change the title of the paper if possible to solve your concerns:
>
> 「Bayesian excess risk analysis for epistemic uncertainty with application to variational inference」.
>
> We add “Bayesian” at the beginning of the title because we would like to clarify that the purpose of this work is not to analyze the standard excess risk, which is often studied in the generalization error analysis. The primarily purpose of this work is to analyze the epistemic uncertainty of the approximation methods used in Bayesian inference, such as MC dropout and deep ensembles. For that purpose, we defined the Bayesian excess risk and analyzed it. We did not analyze the standard excess risk, but we only used the result of the existing analysis for them.
>
> We also remark on the comment, "*my concerned relation about $E_{Y,X}(Y-E_{q(\theta)}f_\theta(X))^2$ and $E_XE_{q(\theta)}(f_\theta(X)-E_{q(\theta)}f_\theta(X))^2)$ was not analyzed in this work. But this is precisely my point: what Sec 3.2 analyzed was not useful (and misleading)*".
>
> As we written in Sec 1, our main interest in this work is to study the widely used epistemic uncertainty (EU) measurements and not to analyze the coverage-problem. Under-coverage problem is an important topic, but it is out-of scope of this paper since it requires to analyze the aleatoric uncertainty and model misspecification problems, which are not included in the definition of Bayesian excess risk..
>
> Our theoretical analysis is meaningful because **1 ) we can understand the behavior of the widely used EU measurements in the uncertainty quantification tasks**, and **2) our analysis leads to our proposed algorithm**.
> 1) For example, we often evaluate the EU measurements by increasing the sample size in the uncertainty quantification tasks. Also, we use the EU measurements conditioned on $x$ for evaluating where in the input space is yet to be learned. Our analysis clarifies that the EU measurements are at least a lower bound of the test error. So, our assumption Eq. (8) implies an optimistic estimate of the test error from the Bayesian viewpoint. Our convergence analysis clarifies that such an optimistic estimate of the error converges at least at the same rate as the test error.
> We remark that our analysis is conditioned on $x$. In this way, it can be used to understand the behavior of the uncertainty quantification tasks.
>
> 2) From the practical viewpoint, we obtain the approximate posterior distributions by variational inference (VI) including the alpha-divergence variant and evaluate its posterior variance. So, we would like to know how the posterior variance differs depending on the choice of different approximation methods. However, since the posterior variance does not appear explicitly in the objective function of approximation methods, that dependency is unclear. Our analysis in Sec 3.3 clarifies this point and leads to our algorithm that directly regularizes the posterior variance.
>
>
> #### **2 ) The motivation of the algorithm was unsatisfactory**
> Following your advice, we will add the following sentences at the beginning of Sec 3.3 so that the motivation and discussion settings in Sec 3.3 become clearer.
>
> (New sentences)
> ```
> In this section, we show how the widely used EU measurements shows different behaviors dependent on the choice of the approximation methods of the Bayesian posterior distribution using theories developed in Sec 3.2.
> It is known that the standard VI, which minimizes the KL divergence often underestimates the EU, while VI based on the alpha-divergence alleviates this issue. This has been explained qualitatively from the mode-seeking and mode-covering property of divergences in the parameter space. We show that this can be qualitatively explained by focusing on the objective functions of those VIs using the decomposition of the test error into the prediction risk and BER developed in Sec 3.2. We will show that our explanation leads to a natural extension of VI that directly controls BER.
> ```

---

> > ### Comment · Reviewer_6WHt · 2022-12-13
> > **Thank you for your response, but it did not address my concerns**
> >
> > It appears that there are still major misunderstandings around my original review, and the updates.  Unfortunately, I'm not quite sure if I could rephrase them for better clarity, and I won't be able to respond to further comments in a timely manner (the author-reviewer discussion period has long ended, although other forms of discussion may still be ongoing).  Perhaps the following questions might help to clarify:
> >
> > - You stated that
> >
> > > our main interest in this work is to study the widely used epistemic uncertainty (EU) measurements and not to analyze the coverage-problem.
> >
> > How could the epistemic uncertainty be analyzed without studying *any* coverage problem? [^1]
> >
> > - Your theorems in Sec 3.2 and the motivating discussions in Sec 3.3 applies to both DNN models with imperfect variational families, and low-dimensional (regular) parametric models with the exact posterior. Do you think there are issues with the latter scenario that need to be addressed?
> >
> > [^1]: As we have discussed, there are many different types of coverage problems. Analysis of EU measurements may also lead to crude statements that only applies to scaled credible sets, as is common in the nonparametric literature.

---

### Official Review · Reviewer_1SWx · 2022-10-26

**Confidence:** 3
**Correctness:** 4
**Technical Novelty And Significance:** 4
**Empirical Novelty And Significance:** 4
**Recommendation:** 6

**Clarity, Quality, Novelty And Reproducibility:**

This paper is overall well-written. It makes a reasonable attempt to provide sufficient background for the readers to understand the proposed methods. The theoretical contribution is novel to me.

**Strength And Weaknesses:**

This work focus on excess risk analysis under approximate Bayesian inference instead of an exact setting as in existing work, which is a more practical setting and seems novel to me. The main theoretical contributions are Thm 1, Thm 2 and Thm 3 where Thm 1 connects Bayesian excessive risk under log loss with approximate mutual information and the one under squared loss with variance and Thm 2&3 upper bounds the summation of prediction excess risk and Bayesian excess risk with convergence analysis.
are nicely explained and seem technically sound.

Still, here are a few concerns:
- p^q is not defined in Eq 4.
- I'm confused by Remark 2. What do you mean by "misspecify noise function"? Can you explain it in a more formal way?
- In Fig 1 and Fig 2, I wonder which N are chosen? From the curves, it seems that the N are not uniformly chosen and also the curves are highly non-smooth. The author might what to try with more N to make some smooth curves that clearly shows the relationship among the three quantities.
- The empirical evaluation would benefit from including more baselines other than particle VI. For example, to include Bayesian Dropout methods including MC dropout, variational dropout, ensembles, Langevin MCMC etc...

**Summary Of The Paper:**

This work explores the relationship among excess risk, widely used epistemic uncertainty and generalization error with convergence analysis in the setting of approximate Bayesian inference. Empirical evaluations on synthetic datasets are carried out to confirm the theoretical results. Further evaluations on UCI datasets and contextual bandit tasks are presented to compare the proposed approach with existing baselines.

**Summary Of The Review:**

Despite some minor confusion and some potential improvement for the empirical evaluations, the contributions in this work seems solid and novel to me.

---

> ### Author Response · Authors · 2022-11-16
> **Answers to concerns**
>
> Thank you for the comments. We will answer each comment below:
> ## p^q is not defined in Eq 4.
> Ans)p^q is defined just after Eq (3), which implies the approximate predictive distribution.
>
> ## Concerns about Remark 2
> Ans)We update Remark 2 in the following way： When we use a flexible model, such as a deep neural network for $f_\theta(x)$, assumption $E_{\nu(Y|x)}[Y|x]=f_{\theta^* }(x)$ holds even when $\nu(Y|x)\neq p(y|x,\theta^*)$, which means we misspecify the noise function in regression.
>
> This means that we only need to correctly specify the “average” of the conditional distribution of $Y$ given $x$, and we do not need to care about the other properties of the distribution of $Y$. In the regression task, $E_{\nu(Y|x)}[Y|x]=f_{\theta^*}(x)$　implies that the regression function is well specified and $\nu(Y|x)\neq p(y|x,\theta^*)$　means that the noise function is misspeficied.
>
> For example, assume that $\nu(y|x)$ is the multi-modal distribution and $p(y|x,\theta)=N(y|f_\theta(x),\sigma^2)$. Then Theorem 1 still holds if average $E_{\nu(Y|x)} [Y]$ is correctly specified by $f_\theta(x)$. However the distribution of p(y|x,\theta) and $\nu(y|x)$ are clearly different.
>
> ## Fig1 and Fig 2 should be smoothed
> Ans)In Fig 1, we tried $N=[500,1000, 3000,5000,8000,10000]$ in the original paper. We increased the number of points to $N=[100,500,1000,2000,4000,5000,6000,7000,8000,9000,10000]$ so that the curves became smooth. We updated Figure 1 in the paper.
>
> In Fig 2, we tried $N=[5000,10000,15000,20000,25000,28000]$. We are trying to increase the number of points, but this takes a  long time since this uses real datasets and the network is U-Net. So currently, we are running the experiment, and we will replace Figure 2 after finishing the experiment.
>
> ## Include other baselines
> Ans)We included the performance of Bayesian dropout, BBBP, and alpha divergence minimization as baselines in Appendix H.
> The results are adapted from [1]. Our method outperformed compared to them.
>
> [1] Deep Bayesian Bandits Showdown, Riquelme, et al., ICLR 2018.

---

> > ### Comment · Reviewer_1SWx · 2022-12-14
> > **thanks for clarifications**
> >
> > I echo the concern raised by Reviewer 6WHt, that the excess risk studied in this work is the excess risk of the stochastic predictor derived from Gibbs posterior instead of that of the deterministic predictor approximating the true Bayes predictor, is critical and since the authors failed to address the concern, this might indicate that this work would require non-trivial modification for it to be qualified for being published and thus I adjust my score accordingly.

---

### Official Review · Reviewer_15vK · 2022-10-28

**Confidence:** 3
**Correctness:** 4
**Technical Novelty And Significance:** 3
**Empirical Novelty And Significance:** 3
**Recommendation:** 8

**Clarity, Quality, Novelty And Reproducibility:**

This is highly original work that builds very nicely off of Xu and
Raginsky's recent work on excess risk. The paper is challenging to
follow due to the theoretical nature of the contributions. It would
also benefit from better signposting where the different versions of
excess risk are introduced and the reason why becomes clearer earlier
in the paper.

This work is of high quality and significance. Due to the theoretical
nature of the work, reproducibility is less of a concern.


**Strength And Weaknesses:**

This paper solves a pressing problem in understanding epistemic uncertainty (EU) and connects it very
well to the more traditional methods of practically estimating EU. The paper offers novel and useful
theoretical contributions which are then supported by demonstratng a regularization strategy and several
numeric experiments. The work though would be valuable even without the additional validation.

The main weakness is some of the contributions could be better signposted in the writing.


**Summary Of The Paper:**

This paper derives PAC-Bayes bounds for epistemic uncertainty in terms of excess risk. A new quantity
is defined called Bayesian Excess Risk which captures the epistemic uncertainty when the true
model is known.


**Summary Of The Review:**

This is interesting and useful theoretical analysis of bounding epistemic uncertainty.

---

> ### Author Response · Authors · 2022-11-16
> **Additional summary of the settings**
>
> Thank you for the comments. Following your suggestion about writing, we added the summary of the settings in Appendix B and added following explanation at the beginning of Sec.2: "In Appendix B, we show summary of the settings."

---

### Decision · Program_Chairs · 2023-01-20

**Decision:**

Reject

**Justification For Why Not Higher Score:**

Concerns over the message communicated by the theoretical theorems are not addressed.

**Justification For Why Not Lower Score:**

N/A

**Metareview: Summary, Strengths And Weaknesses:**

This works studies potential PAC-Bayesian like guarantees for epistemic uncertainty estimates. The approach is to define "Bayesian Excess Risk" as to capture epistemic uncertainty, and the authors provided a number of theoretical results to analyse this quantity.

Reviewers agree that the problem that the paper intended to study is interesting, and the technical approach of extending Xu and Raginsky's recent work is original. They have no problem with the technical correctness of the proofs.

However, concerns have been raised about the interpretation of the theoretical results. In particular, one reviewer pointed out that the theoretical results in section 3.2 can be misleading, as the reviewer and the authors cannot agree on the definition of excess risk.

After a brief read of the paper, I personally think that, while the theoretical technical results (in terms of proofs) are indeed correct, the quantities that the authors studied are not exactly the same as what Bayesian deep learning papers often report in experiments. Indeed in BDL practice for regression problems the experimental results we report (e.g., posterior predictive mean and uncertainty estimates) are computed in the way as Reviewer 6WHt pointed out. Therefore, this means it is not straightforward to directly connect the theoretical results to practical observations. E.g., the authors interpret Thm 2's indication as "consistent with the well-known result that the variance of the predictor often underestimates EU" and cited the Deep Ensemble paper, while to the best of my knowledge, there's no explicit target that allows us to directly evaluate under- or over-estimate of EU in a quantitative sense, and most of the (frequentist) calibration evaluations do not separate EU and aleatoric uncertainty.

In summary, while this paper presented very interesting theoretical tools towards studying epistemic uncertainty estimates, due to concerns of the misleading message connecting the theory and practice, I decide that the paper cannot be accepted in its current form. I encourage the authors to submit this work to another venue, by better clarification of the message of connecting theory with practice.

**Summary Of Ac-Reviewer Meeting:**

After discussions, the main concern appears to be, while the theoretical results are technically speaking correct, the authors seem to have studied the wrong quantities, and made misleading claims based on their theorems. Lengthy author feedback and discussions doesn't seem to resolve this issue. None of the reviewers would like to champion for acceptance.